# The Impact of Targeted Therapies on Red Blood Cell Aggregation in Patients with Chronic Lymphocytic Leukemia Evaluated Using Software Image Flow Analysis

**DOI:** 10.3390/mi16010095

**Published:** 2025-01-15

**Authors:** Anika Alexandrova-Watanabe, Emilia Abadjieva, Lidia Gartcheva, Ariana Langari, Miroslava Ivanova, Margarita Guenova, Tihomir Tiankov, Velichka Strijkova, Sashka Krumova, Svetla Todinova

**Affiliations:** 1Institute of Mechanics, Bulgarian Academy of Sciences, “Acad. G. Bontchev” Str. 4, 1113 Sofia, Bulgaria; anikaalexandrova@abv.bg (A.A.-W.); abadjieva@gmail.com (E.A.); tiho_bg@abv.bg (T.T.); 2Center of Competence at Mechatronics and Clean Technologies—MIRACle, “Acad. G. Bontchev” Str. 4, 1113 Sofia, Bulgaria; arianalangari@abv.bg; 3National Specialized Hospital for Active Treating of Hematological Diseases, Zdrave Str. 2, 1756 Sofia, Bulgaria; l.garcheva@hematology.bg (L.G.); m.genova@hematology.bg (M.G.); 4Institute of Biophysics and Biomedical Engineering, Bulgarian Academy of Sciences, “Acad. G. Bontchev” Str. 21, 1113 Sofia, Bulgaria; miroslava.ilieva.ivanova@gmail.com (M.I.); sashka.b.krumova@gmail.com (S.K.); 5Institute of Optical Materials and Technologies “Acad. Yordan Malinovski”, Bulgarian Academy of Sciences, “Acad. G. Bontchev” Str. 109, 1113 Sofia, Bulgaria; vily@iomt.bas.bg

**Keywords:** chronic lymphocytic leukemia, red blood cell aggregation, microfluidic device, software image flow analysis, red blood cell morphology, Obinutuzumab, Venetoclax, Ibrutinib

## Abstract

Chronic lymphocytic leukemia (CLL), the most common type of leukemia, remains incurable with conventional therapy. Despite advances in therapies targeting Bruton’s tyrosine kinase and anti-apoptotic protein BCL-2, little is known about their effect on red blood cell (RBC) aggregation in blood flow. In this study, we applied a microfluidic device and a newly developed Software Image Flow Analysis to assess the extent of RBC aggregation in CLL patients and to elucidate the hemorheological effects of the commonly applied therapeutics Obinutuzumab/Venetoclax and Ibrutinib. The results revealed that, in RBC samples from untreated CLL patients, complex 3D clusters of large RBC aggregates are formed, and their number is significantly increased compared to healthy control samples. The application of the Obinutuzumab/Venetoclax combination did not affect this aspect of RBCs’ rheological behavior. In contrast, targeted therapy with Ibrutinib preserves the aggregation state of CLL RBCs to levels seen in healthy controls, demonstrating that Ibrutinib mitigates the alterations in the rheological properties of RBCs associated with CLL. Our findings highlight the alterations in RBC aggregation in CLL and the impact of different targeted therapies on RBCs’ rheological properties, which is critical for predicting the potential complications and side effects of CLL treatments, particularly concerning blood flow dynamics.

## 1. Introduction

Chronic lymphocytic leukemia (CLL) is the most common indolent lymphoproliferative neoplasm in adults, characterized by the accumulation of morphologically mature but immune-incompetent lymphocytes in the blood, bone marrow, lymph nodes, and spleen [1]. The etiology of CLL is not fully understood; however, genetic factors, oncogenic viruses, exposure to certain toxic substances, and radiation are among the well-established triggering factors [2]. A major hallmark of the disease is the apoptotic resistance of CLL B cells, which seems to be mediated by increased levels of specific anti-apoptotic proteins such as B-cell lymphoma 2 (Bcl-2). On the other hand, changes in the microenvironment and the resulting extrinsic mis-signaling additionally contribute to uncontrolled cell proliferation [3].

CLL is a heterogeneous disease that requires immediate therapy after diagnosis in some patients, while others may live for decades without treatment. Conventional CLL treatment is supplemented with two types of anti-CD-20 (surface phosphoprotein expressed on most malignant B cells) monoclonal antibodies (mAbs), such as rituximab, and Obinutuzumab, which can completely replace chemotherapy. Type I mAbs (e.g., rituximab) are regarded as the most potent antibodies because of their ability to mediate complement-dependent cytotoxicity by engaging Fc receptors on immune effector cells (natural killer cells and macrophages). Type II mAbs (e.g., Obinutuzumab) have lower complement-dependent cytotoxicity than rituximab but greater antibody-dependent cellular cytotoxicity. Obinutuzumab can directly induce the death of CLL cells, although the exact mechanism through which this occurs is still under investigation [4].

Targeted therapy with anti-CD20 mAbs is often used in combination with Bcl-2 inhibitors—a promising strategy to either restore the normal apoptotic process in cancer cells or make them more susceptible to conventional chemotherapy [5]. Over the last decade, several inhibitors of the Bcl-2 family have been developed. One of the most prominent examples is Venetoclax, a selective Bcl-2 inhibitor that induces apoptotic cascade in CLL cells.

Ibrutinib is a first-of-its-kind antineoplastic agent that irreversibly binds to cysteine residue (Cys481) in the active site of Bruton’s Tyrosine Kinase (BTK), leading to its inhibition and thus blocking the proliferation and survival of malignant B cells [6].

These new targeted therapies were shown to have strong anti-cancer activity and have significantly improved the clinical outcomes of patients with CLL. However, there are still unresolved issues regarding the occurrence of side effects associated with bleeding, atrial fibrillation, the development of hypertension, and further major cardiovascular events. These concerns may be, at least in part, attributed to downstream off-target effects, which can be linked to the impaired microcirculation and altered rheological properties of RBCs [7,8]. Indeed, alterations in endothelial behavior have been observed in patients with lymphoproliferative disorders, including CLL [9,10]. To shed light on this problem in the present study, we characterized RBCs’ rheology in CLL.

The rheological behavior of blood is heavily influenced by the interaction of RBCs with each other, along with their interaction with the blood plasma and the vessel walls. Under normal conditions, at a low shear rate (<10 s^−1^) or in stasis (i.e., in the large vessels), RBCs form a stable yet flexible rouleaux that promotes an increase in the apparent viscosity of whole blood. The rouleaux represents the reversible structures of two- or three-dimensional packages of erythrocytes [11]. Rouleaux formation has the greatest influence on the shear rate dependence of blood viscosity. RBC aggregation depends on extrinsic factors, such as the local shear rate, hematocrit, plasma levels of high-molecular-weight proteins (with fibrinogen being the most valuable), and intrinsic cell properties, including membrane surface charge and fluidity and cell deformability. Increased RBC aggregation and reduced deformability may influence the viscosity of blood and impair peripheral tissue flow resistance and oxygen delivery [12,13].

The dynamics of the aggregation process in the blood is crucial for understanding the rheological properties of RBCs and blood flow behavior under different pathological conditions. In some diseases, such as diabetes mellitus, cancer, and preeclampsia, as well as in various infection and inflammation conditions, an increase in RBC aggregation (hyper aggregation) appears as a non-specific indicator [14,15,16]. The aggregates of more rigid and less deformable RBCs tend to be very robust and are difficult to disperse through increasing the shear rates. For example, RBCs in CLL samples have been shown to appear thinner, with a loose appearance and low density, suggesting an alteration in RBC aggregation in the blood [17].

The last decade witnessed remarkable progress in the development of microfluidic devices, particularly for studies of blood rheology. These devices offer several key advantages over traditional macroscopic rheometers, including the use of smaller sample volumes, achievement of higher shear rates, and creation of cost-effective, disposable devices with precise shear rates that mimic physiological conditions. These devices can also be combined with microscopy or other optical techniques to create a microfluidic system [18,19]. An important advantage of modern microfluidic systems is the dimensions of the microfluidic channels, which can be considered a model of blood vessels [20]. The microfluidic systems demonstrate high sensitivity in detecting small changes in RBC properties [21].

Microfluidic devices, combined with image processing and analysis techniques, are often used to accurately evaluate parameters that characterize RBC aggregation [15,22,23]. Foresto et al. (2000) introduced the aggregate shape parameter as a way of quantifying the shape and morphology of aggregates to assess how these different aggregate shapes might behave under different shear conditions—at stasis and under different shear stresses [23]. By analyzing the microscopic blood images, a new method was proposed using the RBC-free area, hematocrit, and total area of aggregates to assess erythrocyte aggregation [24,25]. Chen et al. (1994) used a computerized image analysis technique to determine the size distribution of RBC aggregates at different shear stresses in flow chambers [26]. Muravyov et al. (2017) developed a specific method called aggregoscopy, which, combined with image processing software, can be used to assess the suspension stability of blood and to quantify RBC aggregation [27]. Mehri et al. (2018) provided an automated methodology for identifying, assessing, and characterizing RBC aggregation at controlled and measurable shear rates in a microfluidic fluid flow shear system using image processing techniques [28]. Recently, we developed an experimental methodology to assess RBC aggregation in the channels of a microfluidic system, combined with a new computer algorithm for the software analysis of flow images at low and high shear rates [15].

The present study aimed to characterize the possible off-target effects induced by conventional CLL-targeted therapy by assessing the extent of RBC aggregation in patients with CLL compared to that of aged-matched healthy individuals. To elucidate the hemorheological effects of the applied therapies, we used a microfluidic system with microchannels as a model of blood vessels. We developed a novel algorithm for Software Image Flow Analysis, a powerful technique for distinguishing different levels of RBC aggregation in the flow, allowing for systematic comparison between study groups and facilitating the interpretation of such complex data. Our results show that the RBC aggregation was significantly higher in CLL samples from untreated patients and patients receiving a combination of Obinutuzumab and Venetoclax than in healthy individuals. In addition, the targeted therapy of CLL patients with Ibrutinib maintained the aggregation of RBCs at the levels seen in healthy controls.

## 2. Materials and Methods

### 2.1. Study Groups and Ethics Statement

Twenty-one patients diagnosed with CLL and admitted to the National Specialized Hospital for Active Treatment of Hematological Diseases, Sofia, Bulgaria, were enrolled in this study. Of these, six were treatment-naïve patients (mean age 57.8 ± 14.0 years), five were patients receiving Ibrutinib (mean age 63.4 ± 10.8 years), and ten were CLL patients receiving a combination of Obinutuzumab/Venetoclax (mean age 69.11 ± 7.8 years). The diagnosis of CLL was determined according to the current guidelines [29,30].

Thirteen healthy controls (mean age 58.0 ± 7.4 years) were also included in the study population. Healthy individuals in the control group had no family history of CLL or other oncohematological diseases.

Treatment regimen:

According to the drug’s SPC (short product characteristics), the CLL patients were treated with Ibrutinib until disease progression or poor tolerability was observed.

The combination of Venetoclax and Obinutuzumab was taken in 12 four-week cycles.

Exclusion criteria: CLL patients with diabetes, renal disease, autoimmune diseases, or hyperlipidemia were not included in this study.

All individuals, both patients and controls, who were involved in the investigation provided written informed consent. The study was approved by the Ethics Committee of the Institute of Biophysics and Biomedical Engineering, Bulgarian Academy of Sciences (approval No 378HД 26/03/2024), and performed according to the Helsinki international ethical standards on human experimentation.

### 2.2. Blood Collection and Sample Preparations

Blood samples were collected via venipuncture into two 6 mL tubes (Vacutainer; Becton Dickinson, and Company, Franklin Lakes, NJ, USA) containing K_2_EDTA. Blood from CLL patients was collected during their hospital visits.

The RBCs were isolated according to the protocol described in [15]. Briefly, after centrifugation of the collected blood (Universal 320 R centrifuge, Hettich, Germany), the supernatant (plasma and white blood cells) was separated and the RBCs were resuspended and washed twice in PBS solution (140 mM NaCl, 2.7 mM KCl, 8 mM Na_2_HPO_4_, 1 mM KH_2_PO_4_). The hematocrit of the prepared RBC suspension was adjusted to 40% (centrifuge Haematokrit 200, Hettich, Germany).

### 2.3. Stimulation of RBC Aggregation

From all blood samples taken from healthy controls, untreated CLL patients, and CLL patients treated with Ibrutinib or a combination of Obinutuzumab/Venetoclax, 10 μL was taken and added to 200 μL of Dextran 70 solution (at a concentration of 4 g/dL) to stimulate RBC aggregation. The final hematocrit of the RBC suspensions diluted in this way was reduced to 2% for all experiments.

### 2.4. Viscosity Measurements

The viscosity of the RBC suspensions diluted in Dextran 70 (according to the method specified in Section 2.3) was defined under steady-state flow conditions using a Brookfield programmable rotational viscometer. DV-II + Pro (Brookfield Engineering Laboratories, Inc., Middleboro, MA, USA). The device was calibrated with water at a temperature of 37 °C. The viscosity of the diluted RBC suspensions in Dextran 70 at a temperature of 37 °C was determined to be 1.12 ± 0.04 mPa.s.

### 2.5. Microfluidic System Description and Experiments

An air-pressure-driven BioFlux microfluidic system (Fluxtion Biosciences, Oakland, CA, USA) was used to study RBC aggregation. A photograph of the microfluidic system used in the experiments of the present study is presented in Appendix A and details of the BioFlux microfluidic system are provided in Appendix A. The system represents a high-quality imaging platform for in-flow rheological analyses. The microfluidic system comprises a BioFlux 200 electro-pneumatic flow control pump, a Lumascope 620 (Etaluma, San Diego, CA, USA) inverted fluorescence microscope, BioFlux microfluidic plates, and a computer configuration with specialized control (Lumaview) and analysis software (Lumaquant 8.8). The RBC aggregation assays were performed in BioFlux 24-well plates 0–20 dyn/cm^2^ with eight microfluidic channels with cross-sectional dimensions of 350 μm width and 75 μm height (Appendix A).

### 2.6. Design of the Experiments

RBC suspension preparation and microfluidic channel setup: The microfluidic channels were filled with 200 μL of each prepared diluted RBC suspension in Dextran 70. The RBC suspension was perfused through the microfluidic channels at a high shear stress of 5 dyn/cm^2^ (corresponding to a shear rate of 446 s^−1^) for 5 min to disperse preexisting aggregates, after which the flow in the channels was suddenly reduced to a low shear stress of 0.1 dyn/cm^2^ (corresponding to a shear rate of 8.9 s^−1^) for 5 min. After the flow was stopped, the formed RBC aggregates were imaged using an inverted microscope operating in the phase contrast mode. A total of 30 s after stopping the flow, at least 5 images were taken every 2 s along the entire visible length of the channel at randomly selected locations.

### 2.7. Algorithm for Software Image Flow Analysis of RBC Aggregates

In the current study, an algorithm was developed to evaluate the RBC aggregates, and a new software program was created based on this algorithm. The program, written in the ImageJ Macro Language, follows a structured workflow to analyze RBC aggregate data and provide software visualizations based on the area of the aggregates. As shown in the illustrated block diagram (Figure 1), the process consists of the following sequential steps.

The first step involves importing the images (as input data) obtained from the BioFlux microfluidic system into Image J 1.54 g. The next phase focuses on image preparation (preprocessing) to enable an accurate image analysis of RBC aggregates. The images are converted into a binary format using a built-in Image J thresholding algorithm, which is essential for separating the RBC aggregates from the background. The thresholding algorithm separates RBC aggregates from the background by converting all pixels above a certain brightness to white and those below to black. In addition, object boundaries are automatically refined to ensure that each RBC aggregate is treated as a single, unified region for accurate analysis.

The next algorithmic step includes configuring the measurements, during which specific parameters, such as the area and number of RBC aggregates, are selected. The area, measured in square micrometers (µm^2^), is the primary metric for segmentation. This parameter is obtained by counting the number of pixels each RBC aggregate occupies, and the result is then converted into real-world size (µm^2^) based on the image’s scale.

The next stage of the software program focuses on the assessment of RBC aggregates, in which the algorithm identifies individual aggregates using an integrated Image J particle analysis function. This includes the following:
Detecting particles based on their area interval (50 µm^2^ and infinity).Adding the detected particles to the Region of Interest (ROI) Manager tool for subsequent analysis and processing.

This methodical approach ensures that data regarding the position and size of each detected RBC aggregate are retained.

The subsequent stage involves color-coding for visualization, in which the RBC aggregates are categorized into predefined size intervals with specific color codes: green 100–330 µm^2^, blue 330–660 µm^2^, cyan 660–1320 µm^2^, magenta 1320–2700 µm^2^, and yellow > 2700 µm^2^.

Each aggregate is processed individually via its selection from the ROI Manager, calculation of its area, and the application of the corresponding color to its region on the image.

In the final step, called Log Data, the algorithm provides a comprehensive summary of the analysis by illustrating key indices: the number of RBC aggregates (*NA*) and the total area of RBC aggregates at each size interval. The data obtained are displayed in the Log Window of ImageJ, allowing for quantitative assessment and further analysis.

### 2.8. RBC Aggregation Parameters

After the Image Flow Analysis, the RBC indices were determined for the 5 different populations defined in Section 2.7. (P1–P5). Two parameters of RBC aggregation were evaluated: the extent of RBC aggregation, obtained using the RBC Aggregation-Area Indicator (*AAI*), and the number of RBC aggregates (*NA*) at a low shear rate (8.9 s^−1^).

The *AAI* for each population represents the RBC aggregation under low-flow conditions, calculated by the formula adapted by [22]:$$AAI=\frac{S_{i}}{S_{V}}$$
where *S_i_* (*i* = 1 ÷ 5, in pixels) is the total sum of the aggregate areas for the five different cell populations (P1, P2, P3, P4, P5) (at a flow of 8.9 s^−1^) and $S_{V}$ (in pixels) is the total observed area of one visual field of the microscope. 

The *AAI*_1_ was calculated for population 1 (P1) RBC aggregates with an area of 100–330 µm^2^, the *AAI*_2_ was calculated for population 2 (P2) 331–666 µm^2^, the *AAI*_3_ was calculated for population 3 (P3) 661–1320 µm^2^, the *AAI*_4_ was calculated for population 4 (P4) 1321–2700 µm^2^, and the *AAI*_5_ calculated was for population 5 (P5) > 2700 µm^2^, respectively. The number of RBC aggregates (*NA*_1_–*NA*_5_) in each population of aggregates in the CLL patients not receiving treatment, CLL patients receiving Ibrutinib or Obinutuzumab/Venetoclax, and healthy controls were evaluated.

### 2.9. Optical Microscopy

The morphological types of RBCs derived from healthy individuals and CLL patients were determined using the optical images of the cells obtained via optical microscopy (3D Optical profiler, Zeta-20, Zeta Instruments, Milpitas, CA, USA). All images were taken with a 50× magnification lens. For RBC smears, RBC suspension (15 μL) diluted with blood plasma (1:1 *volume*:*volume* ratio) was dropped onto a poly-L-lysine slide. The concentration of RBCs in the sample was low enough to prevent a significant number of RBCs from adhering to each other. The drop was then spread evenly over the glass until a thin layer was obtained. The cells deposited on the slides were observed using an optical microscope. The experiments were carried out at room temperature.

### 2.10. Statistics

All data were presented as mean and standard deviation (SD). The Shapiro–Wilk test was used to assess the distribution of the data sets. A non-parametric Wilcoxon test was used to compare data between independent groups. Significant differences were considered at the level of *p* ≤ 0.05. Statistical analyses were performed using the OriginPro 2018 program package.

## 3. Results

### 3.1. Clinical Characteristics of the CLL Patients and Healthy Controls

Twenty-one patients with CLL, 62% of whom were men, were enrolled in this study. Six of the patients had early-stage CLL (Rai stage 0). They were asymptomatic or had mild symptoms that did not warrant treatment. It should be noted that only one patient in this group was newly diagnosed. In the remaining cases, ten patients received a combination of Obinutuzumab/Venetoclax (Rai stage 1–4) and five received Ibrutinib (Rai stage 1–4).

Table 1 presents the clinical and hematological characteristics of the patients. The age range of the study groups was 37 to 73 years (median 66) for untreated patients, and 53 to 80 years (median 73) and 51 to 77 years (median 57.5) for patients treated with Obinutuzumab/Venetoclax and Ibrutinib, respectively. The age of the healthy controls ranged from 45 to 67 years (median 58).

The immunohematological analysis of RBC revealed no pathological anti-erythrocyte antibodies in the patients’ groups. Although the main hematological indices did not differ significantly between the patient and control groups, there were some variations noted within the groups. Notably, low hemoglobin (Hb) levels were defined in four patients receiving Obinutuzumab/Venetoclax treatment and in one untreated patient. Red blood cell counts were also reduced in one patient treated with ibrutinib, three patients treated with Obinutuzumab/Venetoclax, and one untreated patient. The RBC distribution width (RDW) was increased in two cases in both the untreated and Obinutuzumab/Venetoclax treated groups. Lower lymphocytes and WBC counts were found in four patients receiving Obinutuzumab/Venetoclax, whereas these indicators were higher in the untreated group.

### 3.2. Software Image Flow Analysis of RBC Aggregation of Healthy Control and CLL Patient Groups

The aggregation process was followed up in all cases studied at a low shear rate of 8.9 s^−1^ in microfluidic channels of the BioFlux system. Images of RBC aggregates were obtained by examining the flow of cell suspensions in Dextran 70 in microfluidic channels. Figure 2 shows representative RBC aggregation images for the healthy controls, untreated patients, and patients undergoing therapy with Ibrutinib or a combination of Obinutuzumab and Venetoclax. Data analysis revealed that the aggregates were heterogeneous in size and shape. In order to identify the level of RBC aggregation in the observed groups in these images, we developed a specific Software Image Flow Analysis (based on an ImageJ Macro Language software program implemented as a plugin in Image J 1.54 g) that distinguishes them based on their size and shape (for more details, see the Materials and Methods). The software program allowed for the more accurate quantification of the variation in RBC aggregation as well as the differentiation of the type of aggregates. This could lead to a better understanding of the role of aggregation in a variety of pathological conditions.

After applying the Software Image Flow Analysis, the aggregates in the RBC suspensions of the control and CLL groups were classified into several distinct populations (P1–P5) with different dimensions (Figure 3 and Appendix A). The individual unaggregated RBCs are also depicted and defined as population 0 (P0) (Figure 3).

The color-coding registered images resulted in a much clearer visualization of the co-existence and abundance of the different populations in the control and diseased groups (for clarity, the color-coding of the images shown in Figure 2 is presented in Figure 4).

This stratification, assessed by Software Image Flow Analysis, also revealed important details about the structural diversity (rouleaux formations vs. three-dimensional cluster networks) of the aggregates we observed and identified deviations from the healthy state, as discussed in the next paragraph.

### 3.3. RBC Aggregation of Healthy Subjects and CLL Patients, Assessed Using a Microfluidic System

Most of the aggregates in the healthy control group appeared as linear rouleaux-like formations with an area of up to 330 µm^2^ (hereafter referred to as population 1—P1) shaped by a few cells (more than three RBCs) (Figure 2A, Figure 3A and Figure 4A). Another significant portion of the aggregates consisted of relatively short-branched rouleaux (population 2—P2, with a size of 331–660 µm^2^) (Figure 2A, Figure 3B and Figure 4A, Table 2). Quantitative analysis indicated that the area of these two populations accounted for two-thirds of the total area of all aggregates observed in one visual field (Appendix A). The number of RBC aggregates (*NA*) in these two populations of rouleaux formations (100–660 µm^2^) is more than 93% of all aggregates (Appendix A). Medium-sized aggregates with an area of 661–1320 µm^2^ (population 3—P3) and larger aggregates with more complex three-dimensional structures (Population 4—P4, sized 1321–2700 µm^2^) were also visible. They were less abundant (<10%), occupying about 24% of the total area of all RBC aggregates (Figure 2A and Figure 3C,D, Appendix A). It should be noted that the aggregates from P4 were found in only half of the samples of the control group, highlighting the variability in the structural complexity of RBC aggregates in healthy individuals.

Image analysis revealed an increased RBC aggregation in untreated CLL patients compared to cells from healthy individuals. In contrast to the controls, the small rouleaux (populations 1 and 2) were significantly reduced in untreated CLL RBC suspensions, accounting for only 33% of the total aggregate area. The number of RBC aggregates from P1 and P2 was 72% of all rouleau-like formations (Table 2). This reduction in the area and number of aggregates was offset by an increase in larger 3D clusters (P3 and P4), with sizes between 661–1320 µm^2^ and 1321–2700 µm^2^ (Figure 2B and Figure 4B). P3 and P4 occupied approximately 44% of the total area, and their number represented 21% of all RBC aggregates (Table 2 and Appendix A). In addition, a distinctive characteristic of RBC behavior in untreated patients was the presence of huge 3D cluster formations exceeding 2700 µm^2^. They formed a dense network of aggregates, occupying more than 24% of the total area, and 4% of the total number (Appendix A).

The individual unaggregated RBCs appeared more abundant in the healthy control samples than in the patient samples (Figure 4).

### 3.4. Effect of In Vivo Target Therapies on RBC Aggregation

The effect of targeted therapies (specifically, BTK inhibitors (Ibrutinib) and BCL-2 inhibitors (Venetoclax) in combination with the monoclonal antibody Obinutuzumab) on RBC aggregation was also investigated. Image flow analysis indicated that treatment with Obinutuzumab/Venetoclax had a negligible impact on cell aggregation related to untreated CLL cases (Figure 2C and Figure 4C vs. Figure 2B and Figure 4B). The small rouleaux in these samples increased compared to the untreated CLL group (Table 2). The P1 and P2 rouleau formations (<660 µm^2^) in CLL patients receiving Obinutuzumab/Venetoclax represented 47% of the total aggregate area, but their number was close to 80% of all RBC aggregates (Appendix A). The 3D aggregate clusters (P3–P5) were slightly reduced as compared to the untreated cases (Figure 2, Appendix A). P3 and P4 occupied 37% of the area of all RBC aggregates, before their number decreased and they represented 15% of all aggregates. The aggregate networks (>2701 µm^2^) occupied 15% of the total area, but they were very few in number—2% of all RBC aggregates (Table 2 and Appendix A).

Our results showed that treatment with the BTK inhibitor Ibrutinib significantly improved the rheological properties of RBC compared to those observed in healthy individuals (Figure 2D and Figure 4D). Specifically, the percentage of the two populations of small rouleau-like aggregates (P1 and P2) was about 70% of the total aggregate size and their *NA* was about 90%, which is close to those of the control group (Appendix A).

There was a marked reduction in the large aggregates with an area between 1321 and 2700 µm^2^ (P3 and P4), and no aggregates greater than 2700 µm^2^ (P5) were detected, resembling the aggregates present in the control group (Figure 2D, Appendix A).

### 3.5. Aggregation Indices Calculated for the Aggregation State of RBCs from Healthy Subjects and CLL Patients

We further assessed the most relevant and informative rheological parameter, the RBC Aggregation-Area Indicator (*AAI*), which quantitatively reflects the degree of aggregation as the ratio of the area of all aggregates to the observed area in one visual field of the microscope. In this study, we applied a new approach to highlight patient-specific features of aggregate formation, particularly in large cluster networks. To achieve this, we calculated the RBC Aggregation–Area Indicator separately for the five defined distinct population ranges of rouleaux formations (P1 and P2) and 3D clusters from aggregates (P3, P4, and P5), as previously described and detailed in Figure 3 (see also Appendix A). The results are summarized in Table 2 and graphically presented as a histogram in Figure 5.

This method provided a more precise understanding of how RBC aggregation varies among patients and under different treatment conditions. In particular, the Aggregation-Area Indicator *AAI*_1_ was the highest for the linear rouleau (*AAI*_1_) populations in the control group and gradually decreased with an increase in aggregate area. In the untreated patient group and the Obinutuzumab/Venetoclax-treated patient group, *AAI*_1_ was significantly reduced by approximately 50% compared to the values observed in healthy controls. Additionally, *AAI*_2_ decreased by more than 30% in both patient groups relative to the control values (Figure 5). The *AAI*_3_ was very similar for healthy controls and for both the group of patients with CLL who were not treated and those who were treated with Obinutuzumab/Venetoclax (Table 2).

Importantly, the *AAI*_4_ for the untreated patient and Obinutuzumab/Venetoclax-treated patient groups was significantly higher than that of the healthy controls, indicating a marked alteration in the aggregation behavior of larger aggregates in these patient groups, suggesting that this treatment and disease state profoundly affected RBC aggregation dynamics. However, it is noteworthy that the number of the largest aggregates in the Obinutuzumab/Venetoclax-treated group was two times lower than in the untreated patients, while the numbers for all other population sizes were comparable (Figure 5, Table 2). It should be noted that *AAI*_5_ and the number of aggregates with an area >2700 µm^2^ had the highest values for these two patient groups (Figure 5, Table 2).

The RBC aggregation indices and the number of aggregates across the different aggregates’ populations for the group of patients receiving Ibrutinib were not significantly different from those determined for healthy cells (Figure 4, Table 2 and Appendix A). Notably, in contrast to the control group, aggregates of P4 were present in all patient cases, indicating a change in the structural dynamics of RBC aggregation that may be related to the disease state. This indicates that the RBC aggregation patterns in the group of patients receiving Ibrutinib were similar to those observed in healthy individuals, suggesting that treatment effectively normalized these dynamics.

We also observed that the ratio of unaggregated erythrocytes to the visible field area, calculated for Population 0 (with an area <99 µm^2^), was the highest in the healthy group and CLL patients treated with Ibrutinib (Table 3). A significantly lower value for this parameter was found in untreated CLL patients and those receiving Obinutuzumab/Venetoclax. This decrease in unaggregated RBCs in the latter groups provides further evidence of increased RBC aggregation in these patients.

### 3.6. Morphology of RBCs Derived from Healthy Individuals and CLL Patients

The morphology of freshly isolated RBC from healthy controls, untreated CLL patients, and CLL patients receiving Obinutuzumab/Venetoclax or Ibrutinib treatment was characterized via optical microscopy (representative images are shown in Appendix A). The quantitative results of the different morphological types observed for their distribution in fresh RBC are presented in Table 4. The analysis revealed that the biconcave cells were the dominant type in all groups. However, their proportion was lower in the RBC suspensions of untreated CLL patients and those treated with Obinutuzumab/Venetoclax (*p* < 0.05) compared to the control samples. The proportion of echinocytes and spherocytes in these two patient groups increased significantly compared to the control cells. It should be noted that the ratio of the three morphological cell classes in patients treated with Ibrutinib was similar to the ratio in the control group.

## 4. Discussion

Understanding the effects of RBC aggregation is of great interest in predicting potential complications related to impaired blood flow circulation and possible thromboembolic events and/or overall cardiovascular risk in CLL patients. Recent studies have reported a high incidence of venous thromboembolism (VTE) among CLL patients [31,32]. The increased risk for thrombotic complications is associated with several factors, including abnormal blood coagulation mechanisms, enhanced platelet activation, and/or altered RBC aggregation.

Here, we focused on assessing two main aspects of the aggregation behavior of RBCs: (1) how the disease itself alters this behavior compared to the healthy state and (2) the effect of two targeted therapies (Obinutuzumab/Venetoclax and Ibrutinib). Our work provided new insights into the role of RBC aggregation behavior in off-target effects and microcirculation in CLL, as discussed below.

### 4.1. Alterations in RBCs’ Rheological Properties in Untreated CLL Patients

The presented results demonstrate a significant increase in the RBC aggregation state in untreated CLL patients compared to healthy individuals, which might result from B cell infiltration in the bone marrow and/or persistent inflammation, as discussed below.

Even though CLL primarily affects B lymphocytes, the disease might also significantly impact RBC aggregation through several mechanisms, including altered plasma protein levels, changes in RBC membrane characteristics, and abnormal interactions with malignant B cells and the tumor microenvironment [33]. Since our experiments were performed in a milieu devoid of plasma proteins and other blood components, the impaired rheological properties detected in RBCs from CLL patients are probably due to previous damage obtained in the disease-specific environment. The increased indices, the number of large aggregates (P4), and particularly the appearance of more complex 3D aggregate clusters (P5), which were not observed in the control group, at the expense of reduced small rouleaux structures, provide evidence of an enhanced RBC aggregation state in the untreated patient group. Likewise, it demonstrates that altered RBC aggregation patterns are associated with the presence of a CLL condition. Although the patients in this cohort were classified as being at the RAI 0 stage, our findings indicate that the disease exhibits aggressive behavior regarding RBCs’ biophysical and mechanical properties. The increased area and number of 3D clusters of large aggregates found in CLL patients can significantly affect blood viscosity. This increased aggregation and blood viscosity can lead to increased hydrodynamic resistance in blood vessels at a low shear rate, which, in turn, can contribute to serious cardiovascular and thrombotic complications [34,35].

The interaction between malignant B cells and erythrocytes could be classified as one of the factors to consider when studying the alterations in RBC aggregation. Malignant B-cells infiltrate the bone marrow and disrupt normal hematopoiesis and blood cell production. In fact, anemia is common in many cases of CLL and arises due to several underlying factors, including leukemic infiltration of the bone marrow [36]. This malignancy can lead to the production of autoantibodies, further contributing to anemia. In our study, most of the observed untreated CLL cases were in the RAI 0 stage without indications for anemia. Thus, the altered RBC behavior is not likely to be attributed to this state. Although the patients do not exhibit full-blown anemia at this stage, subtle hematologic abnormalities could still be present. Abnormalities in RBCs, such as changes in their shape, size, or lifespan, might be due to the presence of the malignancy itself rather than overt anemia. In line with this, Bortolato et al. (2021) state that, unlike anemia, leukemia has a different mechanism of action, triggering different biological processes that lead to changes in the erythrocyte membranes and their hemorheological properties [37]. RBCs are highly susceptible to the constantly changing environment in the bloodstream, which can cause significant variations in their morphology or biophysical properties. The altered immune environment in CLL could lead to the accelerated destruction or accelerated aging process of RBCs, resulting in the appearance of damaged or dysfunctional RBCs in circulation. In support of this statement, our results demonstrated the occurrence of morphological alterations in the RBCs of untreated CLL patients, expressed as an increase in the proportion of cells with reduced deformability (i.e., echinocytes and spherocytes) compared to the control group (Table 3).

RBC deformability is largely determined by the elasticity and flexibility of their membrane, which is affected by a complex interplay of factors, including the lipid and protein composition of the membrane, the cytoskeletal integrity, the environmental conditions (such as pH, ion concentrations, etc.), and genetic factors. Diseases and treatments that disrupt any of these factors can significantly impair RBCs’ deformability. Key membrane proteins like spectrin, ankyrin, and band 3 are essential for maintaining the structural integrity of the RBC membrane. For example, it has been shown that a reduction in the number of ankyrin-binding sites in the cytoplasmic domain of band 3 in red blood cells from patients with chronic myeloid leukemia partially disrupts the connection between the cytoskeleton and the membrane [38]. In hereditary spherocytosis, the deficiency or dysfunction of proteins like α-spectrin, β-spectrin, ankyrin, band 3, or protein 4.2 impairs the normal interaction between the RBC membrane and its cytoskeletal network, making the RBCs more fragile and prone to hemolysis. RBC deformability influences the resilience of cells, which is crucial for their proper function in circulation [39,40]. It is also closely linked to the modifications in its protein and lipid membrane under pathological conditions [41]. In this regard, Hon et al. demonstrated changes in RBC membrane fluidity in patients with multiple sclerosis owing to membrane lipid alterations [42]. The degradation of cell deformability also depends on variations in cell shape, which can occur due to various factors, including pathological conditions, aging, and/or the accumulation of non-neutralized reactive oxygen species (ROS) [43]. Sickle cell anemia is often considered the prototype of RBC rheological diseases [33]. Therefore, it would be of great interest to study the CLL RBC membrane’s physical properties in the future.

Another factor, such as the aging of RBCs, can lead to increased aggregation in microcirculation. In their work, Puthumana and colleagues state that aging and increased cell density are associated with dehydration, membrane loss, and other changes leading to smaller, less deformable, and more spherical cells (e.g., echinocytes), and consequently to increased RBC aggregation [44]. In line with this statement, we found a higher proportion of echinocytes and spherocytes in the samples of untreated CLL patients and those receiving Obinutuzumab and Venetoclax, evidencing accelerated RBC aging in these patients, as mentioned previously. Impaired deformability and increased RBC aggregation impede their passage through smaller capillaries, preventing them from delivering oxygen efficiently [40]. Despite the growing evidence of harmful RBC deformability in various pathological conditions, few studies have focused on RBCs’ deformability in hematological diseases. Addressing this, recently, Lee et al. reported that RBC deformability is generally observed in leukemic diseases compared to control cells [45].

Inflammation is another important factor affecting RBC behavior. Chronic inflammation is known to be a key player in the pathophysiology of CLL, as evidenced by the increased production of inflammatory cytokines (such as IL-6, IL-10, IL-8, TNF-α, and IFN-γ) and chemokines, alongside the activation of intracellular pro-inflammatory signaling pathways [46,47,48]. Wu et al. reported that plasma IL-16 levels were significantly elevated in CLL patients at the Rai stage [49]. The nuclear factor–kappa B (NF-κB) pathway was found to be a key regulator of inflammation and is frequently activated in CLL. The activation of NF-κB is often linked to chronic exposure to inflammatory cytokines within the tumor microenvironment [50]. In addition, chemokines like CCL3, CCL4, and CCL5, which are involved in recruiting different immune cell subsets to the tumor microenvironment, help to create a supportive niche for CLL cells by facilitating their interaction with other immune cells, including T lymphocytes, macrophages, and dendritic cells. Additionally, these interactions can lead to the secretion of further inflammatory mediators [51,52].

RBCs are the blood cells most susceptible to oxidation. In a normal state, effective RBC enzymatic antioxidant systems and low-molecular-weight antioxidants protect RBC from oxidative damage. However, the persistent inflammation caused by CLL can alter the equilibrium between oxidation and the cells’ defense mechanisms, leading to changes in RBC membrane properties. The elevated levels of pro-inflammatory cytokines in CLL promote the activation of RBC NADPH oxidase, thereby increasing the production of internal ROS [53]. ROS, in turn, contributes to the aggravation of RBC dysfunction by increasing cell stiffness. Enhanced oxidative stress damages the viscoelastic properties of the RBC membrane, further impairing cell deformability and altering their aggregation [54].

Our finding that the unaggregated RBCs in the control group were higher than those in the patient group suggests that RBCs in untreated CLL patients are less likely to preserve their normal deformability and membrane integrity, impairing their dispersion in contrast to healthy cells.

### 4.2. Effect of Targeted Therapies on RBC Aggregation in CLL

Investigating how targeted therapies like Ibrutinib, Obinutuzumab, and Venetoclax influence RBC membrane properties, plasma viscosity, and cytokine profiles involves understanding the molecular mechanisms behind these drugs’ mode of action, as well as their effects on blood flow, aggregation, and overall hemodynamics. Each of these targeted therapies has distinct mechanisms of action that can affect various components of the blood system but are poorly understood.

Treatment with a combination of Obinutuzumab and Venetoclax did not significantly alter the observed overall increase in RBC aggregates and the RBC morphology as compared to untreated CLL patients. This finding may suggest that the enhanced RBC aggregation in CLL remains unaffected by this combination of therapeutics.

It should be noted that 40% of the recruited CLL patients treated with Obinutuzumab/Venetoclax experienced anemia, which is expected to alter the characteristics of RBCs and potentially affect aggregation. Anemia increases oxidative stress in RBC, promoting reduced cell deformability [55]. However, it is difficult to distinguish the impact of the disease itself through the effect of anemia on RBC morphology and aggregation because both conditions can induce overlapping changes in RBC behavior. Therefore, further research is needed to fully elucidate the mechanisms of action of this combination therapy on RBCs in CLL.

While much of the research on Ibrutinib has focused on its effects on immune cells, some studies and indirect evidence suggest that Ibrutinib could affect other cell types, including RBCs [56,57]. Our results demonstrate the positive effect of this BTK inhibitor in restoring the rheological properties of CLL RBC close to the normal values. The significant reduction in large aggregates, along with the significant reduction in the number and area of large RBC aggregates (P3–P4) to values comparable to those of healthy cells, highlight its potential therapeutic benefits. This demonstrates that treatment with Ibrutinib effectively changes RBCs’ aggregation characteristics, aligning them closely with those of healthy individuals. CLL patients treated with Ibrutinib also showed a high number of unaggregated RBCs, similar to healthy individuals. This indicates that Ibrutinib treatment might help maintain RBCs’ deformability and prevent excessive aggregation, possibly by modulating the inflammatory or immune environment, which could otherwise impair RBC aggregation. The primary target of Ibrutinib is the inhibition of BTK’s enzymatic activity [6]. This, in turn, leads to indirect effects, namely the changes that occur due to the elimination of malignant CLL cells and/or disruption of the interaction between CLL cells and the tumor microenvironment [58]. Niemann et al. (2016) demonstrated that serum levels of key chemokines and inflammatory cytokines decreased significantly in patients on Ibrutinib [59]. As a result, the effects of prolonged oxidative stress were dramatically reduced while the production of newly formed RBCs in the bloodstream was unaffected.

Last but not least, it should be noted that Ibrutinib also targets BTK in platelets, which can alter platelet activation, aggregation, and interaction with other blood components [60]. It has been shown that Ibrutinib inhibited GPVI-induced platelet activation and thrombus formation on surfaces co-coated with vWF [61]. While this mainly affects thrombosis and clot formation, it could also indirectly affect RBC features, as platelets and RBCs interact in the bloodstream [62,63]. Therefore, elucidating details on the mechanism by which Ibrutinib influences RBC aggregation is of interest and will be a subject of our future studies.

Our findings on the rheological changes in red blood cells, including their aggregation–area indicator, reflect their aggregation dynamics in a controlled environment with reduced hematocrit. However, in in vivo conditions, such as those found in whole blood, RBC aggregation is influenced by more complex factors, including variations in hematocrit and plasma protein levels, which can significantly affect aggregation in a more complex way. It is well known that RBC aggregation is particularly sensitive to hematocrit levels, especially when the hematocrit exceeds 20–30% [64]. As mentioned above, the bridging action of high-molecular-weight plasma proteins—such as fibrinogen and albumin—plays a critical role in RBC aggregate formation, particularly in larger veins [65]. Altered protein levels, influenced by various factors, such as inflammation, stress, and chronic diseases (e.g., diabetes, cardiovascular disease, cancer) can exacerbate the aggregation of RBCs and lead to a higher blood viscosity, which affects overall circulation [66,67]. Therefore, RBCs’ aggregation and deformability, and their interaction with plasma proteins, all contribute to blood viscosity and flow. Any changes or disturbances in these factors caused by diseases or treatments can lead to microcirculatory dysfunction and impaired blood flow.

It should also be noted that RBC aggregation increases in the presence of Dextran due to its macromolecular interactions with the RBC membrane, which alter the forces between RBCs, and promote the formation of rouleaux. The extent of aggregation depends on the concentration and molecular weight of Dextran. However, the concentration of Dextran used in our experiments is very low and equal across all samples, meaning its effect is minimal and does not influence the results obtained.

In the present work, we excluded the influence of extrinsic factors, allowing us to specifically assess RBC aggregation based on the intrinsic biophysical characteristics of the RBCs and the changes induced by CLL and its corresponding treatment. Thus, our approach emphasizes the relationship between alterations in the biophysical properties of RBCs and their rheological behavior in disease. Additionally, RBC aggregation can be influenced by substances such as Dextran, a polysaccharide used to mimic blood viscosity and cell behavior. When introduced to a suspension of RBCs, Dextran acts as a macromolecular bridge that promotes RBC aggregation by facilitating intercellular interactions [68].

## 5. Conclusions

In this study, we developed Software Image Flow Analysis to differentiate the effect of different targeted therapies on the rheological properties of RBCs in CLL. Using the Software Image Flow Analysis, we found that RBC aggregation is significantly increased in untreated CLL patients compared to healthy controls.

The targeted therapy of CLL patients with a combination of Obinutuzumab/Venetoclax does not increase RBC aggregation compared to untreated patients. This finding suggests that the mechanisms that drive RBC aggregation in CLL may remain unaffected by this therapy and that it does not directly alter the mechanisms that lead to increased RBC aggregation in CLL.

The targeted therapy of CLL patients with Ibrutinib restores the aggregation state of RBCs to levels seen in healthy controls, suggesting that ibrutinib has a beneficial effect not only on immunologically incompetent lymphocytes but also on the rheological properties of the blood. Specifically, the effect of Ibrutinib may be related to its immunomodulatory activity, which reduces chronic inflammation and cytokine levels—both of which play a role in altering RBC membrane properties and promoting aggregation.

Understanding the changes in RBC aggregation in CLL patients and the impact of the targeted therapies is critical for predicting potential complications, including thromboembolic events and cardiovascular risk. The quantification of RBC aggregation may therefore be a useful tool for monitoring CLL disease progression, improving the process of clinicians’ decision-making.

The Image Flow Analysis Software we developed represents a powerful tool for both basic research and clinical applications. It allows for a more nuanced and comprehensive view into RBC aggregation, consisting of the following:(1)The ability to classify RBC aggregates based on their size and to capture subtle differences in their dimensions, thus providing information on their contribution to various diseases;(2)Differentiation of aggregate types: the ability to distinguish between different aggregate types (such as rouleaux formations vs. three-dimensional cluster networks) offers deeper insight into how aggregation mechanisms might change under pathological conditions.

## Figures and Tables

**Figure 1 micromachines-16-00095-f001:**
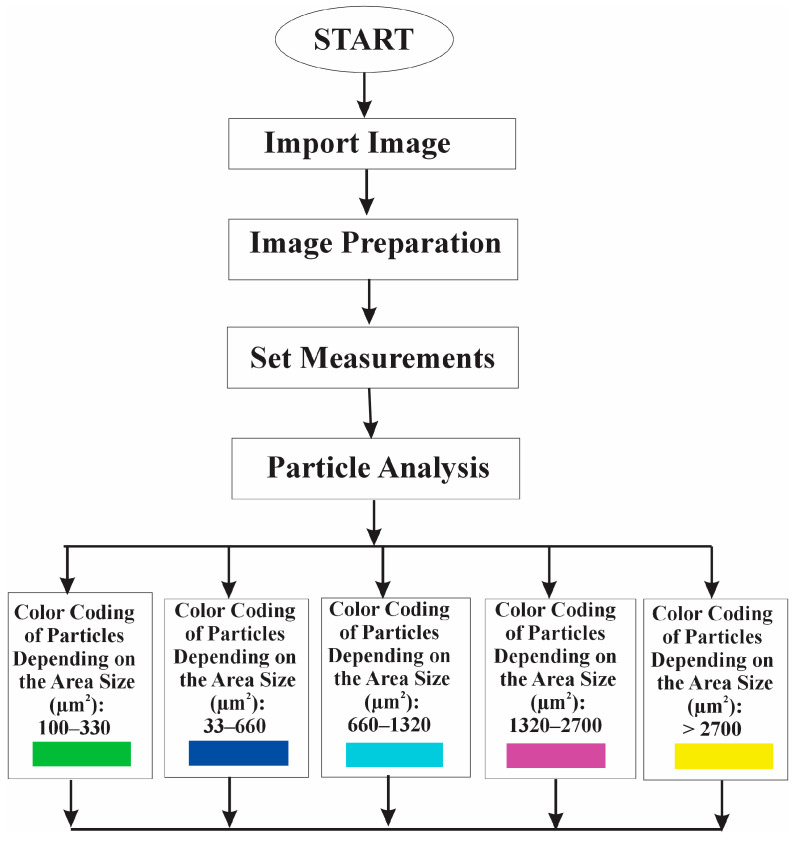
Algorithm diagram for image flow analysis of RBC aggregates.

**Figure 2 micromachines-16-00095-f002:**
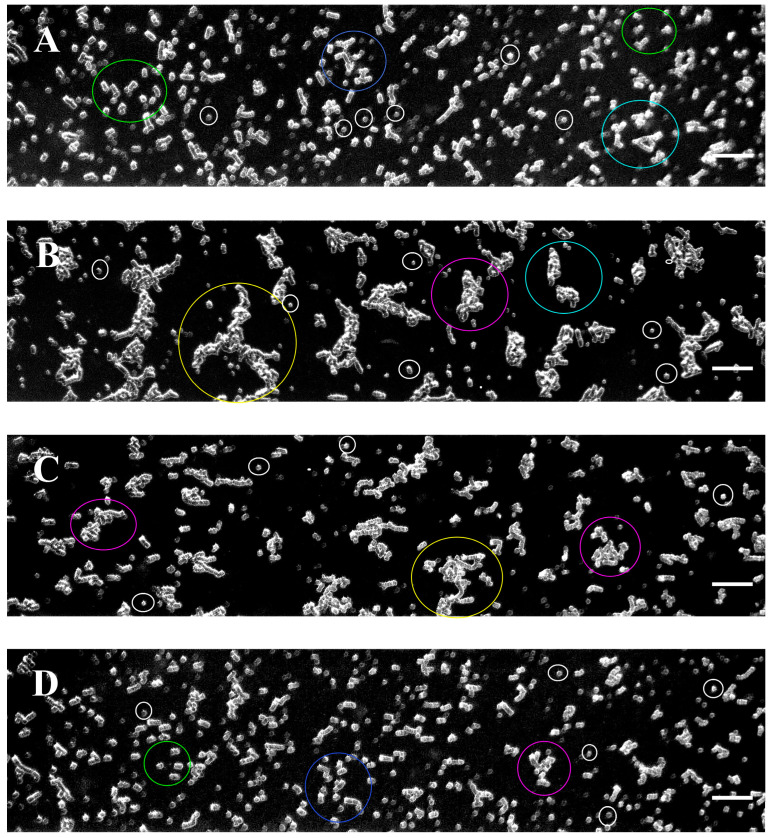
Representative images of the RBC aggregation process obtained with the BioFlux microfluidic system under low-flow conditions (8.9 s^–1^) for (**A**) healthy individuals; (**B**) patients with CLL; (**C**) patients receiving Obinutuzumab/Venetoclax; and (**D**) patients receiving Ibrutinib. Some of the encircled RBC aggregates (examples of the five populations of cell aggregates are depicted in the same colors used in the color-coding) are enlarged in Figure 3. Examples of the individual unaggregated RBCs—population 0—are depicted by a white circle. Scale bar—50 µm. Each image in Figure 2 represents 1600 × 300 pixels^2^.

**Figure 3 micromachines-16-00095-f003:**
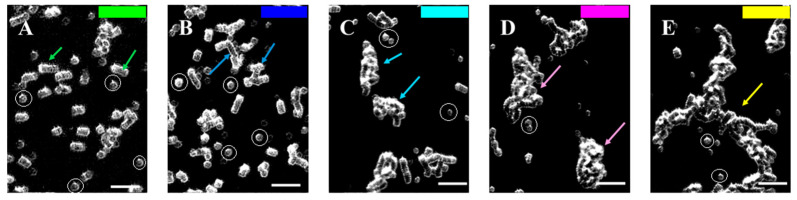
Representative images of the selected individual aggregates, selected from Figure 1 (enclosed by colored circles), shown in increasing order of aggregation, as follows: (**A**) small linear rouleaux (shown with green arrows, 100–330 µm^2^); (**B**) branched rouleaux (shown with blue arrows, 331–660 µm^2^); (**C**) medium 3D aggregates (shown with light blue arrows, 661–1320 µm^2^); (**D**) large 3D aggregates (shown by purple arrows, 1321–2700 µm^2^); and (**E**) aggregate networks (shown with yellow arrows, >2700 µm^2^). The color code for the corresponding RBC aggregate area is presented in the upper right corner of each image. The colored arrows indicate the aggregates with the corresponding area. The individual disaggregated RBCs are encircled with white lines. Scale bar—25 µm.

**Figure 4 micromachines-16-00095-f004:**
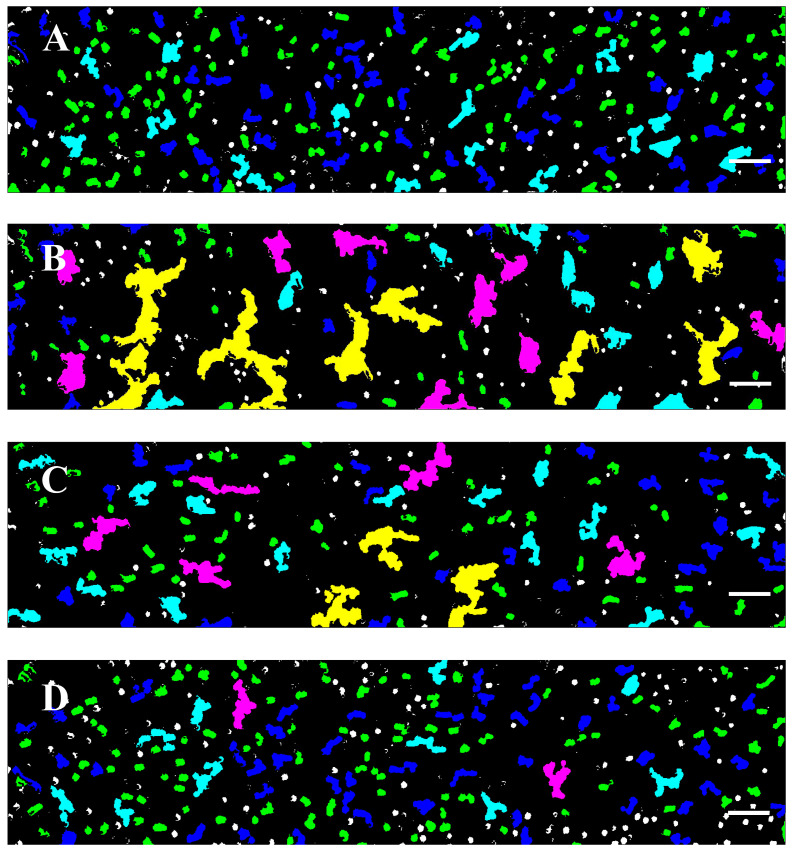
Color-coding of the representative images of the RBC aggregation, presented in Figure 2 based on the developed Software Image Flow Analysis. (**A**) Healthy individuals; (**B**) patients with CLL; (**C**) patients receiving Obinutuzumab/Venetoclax; and (**D**) patients receiving Ibrutinib. Scale bar—50 µm. Aggregated populations 1–5 (as defined in Figure 3) are presented in green, (population 1—P1, 100–330 µm^2^), blue (population 2—P2, 331–660 µm^2^), light blue (population 3—P3, 661–1320 µm^2^), purple (population 4—P4, 1321–2700 µm^2^), and yellow (population 5—P5, (>2701 µm^2^). The individual unaggregated RBCs—population 0, (with area < 99 µm^2^)—are presented as white spots. Each image in Figure 4 represents 1600 × 300 pixels^2^.

**Figure 5 micromachines-16-00095-f005:**
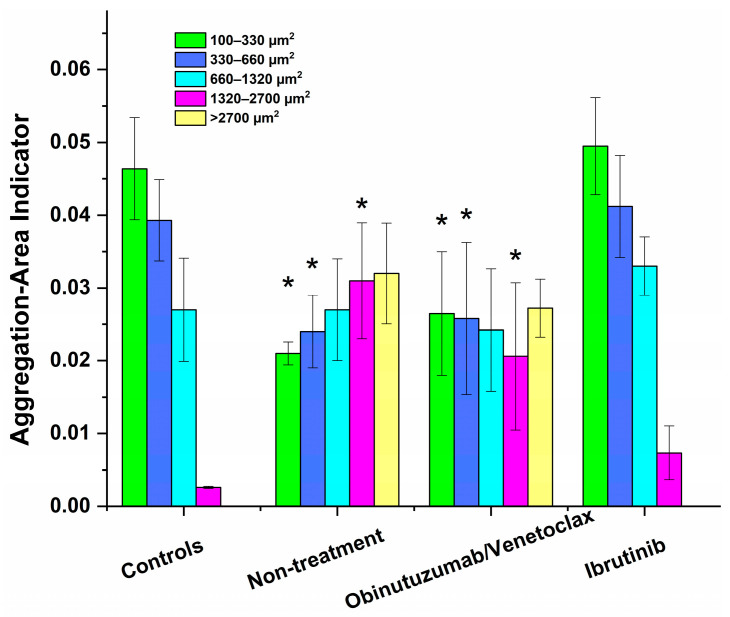
Histogram of Aggregation-Area Indicator values calculated based on the applied Software Image Flow Analysis for the five distinct population ranges of rouleaux formations and 3D clusters, i.e., P1 (100–330 µm^2^), P2 (331–666 µm^2^), P3 (661–1320 µm^2^), P4 (1321–2700 µm^2^), and P5 (>2700 µm^2^), for the untreated patients, patients receiving Ibrutinib or Obinutuzumab/Venetoclax, and healthy controls. * Indicates statistically significant difference (*p* < 0.05) in the values of Aggregation-Area Indicator for CLL patient groups compared with the control values.

**Table 1 micromachines-16-00095-t001:** Clinical data (age, gender, and laboratory indices (RBC count; hemoglobin, Hb; hematocrit, Ht; mean corpuscular volume, MCV; mean corpuscular hemoglobin, MCH; mean corpuscular hemoglobin concentration, MCHC; red blood cell distribution width, RDW; total bilirubin; white blood cell (WBC), lymphocytes; and platelet count of the healthy controls and CLL patients.

Characteristics	Groups				
	Reference Values	Healthy Controls (n = 13)	CLL Patients Without Treatment (n = 6)	CLL Patients Receiving Obinutuzumab/ Venetoclax (n = 10)	CLL Patients Receiving Ibrutinib Treatment (n = 5)
age (years)	-	58.00 ± 7.4	57.75 ± 14.0	69.11 ± 7.8	63.4 ± 10.8
gender (F/M)		8/5	2/4	4/6	2/3
RBC count (T/L)	4.60–6.20	4.97 ± 0.23	4.97 ± 0.41	4.62 ± 0.45	4.80 ± 0.20
Hb (g/L)	140.00–180.00	161.40 ± 8.09	150.50 ± 12.82	136.70 ± 12.94 *	141.80 ± 5.84
Ht (L/L)	0.40–0.54	0.48 ± 0.01	0.45 ± 0.02	0.41 ± 0.03	0.43 ± 0.02
MCV (fL)	80.00–95.00	89.10 ± 3.74	90.70 ± 4.69	88.99 ± 5.96	89.74 ± 1.81
MCH (Pg/L)	27.00–32.00	30.55 ± 1.33	30.30 ± 1.46	29.69 ± 2.39	29.54 ± 1.18
MCHC (g/L)	320.00–360.00	344.75 ± 4.15	334.50 ± 11.67	333.40 ± 13.93	329.60 ± 17.51
RDW %	11.60–14.80	13.88 ± 0.80	14.60 ± 1.54	14.26 ± 1.06	13.90 ± 0.57
WBC	3.50–10.50	6.3 ± 1.1	11.7 ÷ 157.5 *	3.88 ± 1.00	6.43 ± 1.34
total bilirubin (umol/L)	3.40–20.50	15.73 ± 11.46	15.73 ± 11.46	17.98 ± 14.04	19.33 ± 11.06
lymphocytes (ABS)	1.10–3.80	1.91 ± 0.17	6.70 ÷ 144.28 *	1.31 ± 0.49	2.08 ± 1.19
platelet count ×10^9^/L	142.00–440.00	301.67 ± 92.63	208.25 ± 31.27	170.70 ± 43.19	166.80 ± 20.74

* Indicates statistically significant difference (*p* < 0.05) in the values of laboratory indices for CLL patient groups compared with the control values; Note: The lowest and highest values for white blood cells (WBC) and lymphocytes in the untreated group are provided.

**Table 2 micromachines-16-00095-t002:** Red blood cell Aggregation-Area Indicators (*AAI*_1_–*AAI*_5_) calculated for aggregates with an area of 100–330 µm^2^ (*AAI*_1_), 331–666 µm^2^ (*AAI*_2_), 661–1320 µm^2^ (*AAI*_3_), 1321–2700 µm^2^ (*AAI*_4_), and > 2700 µm^2^ (*AAI*_5_), respectively; number of RBC aggregates (*NA*_1_–*NA*_5_) at low shear rates in the patients not receiving treatment, patients receiving Ibrutinib or Obinutuzumab/Venetoclax, and healthy controls; mean values and SD.

Parameters	Groups			
	Healthy Controls	CLL Patients Without Treatment	CLL Patients Receiving Obinutuzumab/ Venetoclax	CLL Patients Receiving Ibrutinib Treatment
*AAI* _1_	0.046 ± 0.007	0.022 ± 0.009 *	0.026 ± 0.01 *	0.054 ± 0.011 ^♦^
*NA* _1_	106 ± 10.6	45.5 ± 23 *	58.3 ± 24.3 *	119 ± 38.5 ^♦^
*AAI* _2_	0.039 ± 0.008	0.024 ± 0.006 *	0.026 ± 0.009 *	0.040 ± 0.007 ^♦^
*NA* _2_	34.7 ± 7.4	20.5 ± 6.2 *	22.2 ± 9.8 *	35 ± 7.8 ^♦^
*AAI* _3_	0.027 ± 0.01	0.027 ± 0.008	0.024 ± 0.008	0.030 ± 0.006
*NA* _3_	10.3 ± 3.2	12.5 ± 3.6	10.6 ± 4.1	13.8 ± 3.0
*AAI* _4_	0.0026 ± 0.0002	0.031 ± 0.008 *	0.021 ± 0.009 *	0.007 ± 0.003
*NA* _4_	1.0 ± 0.7	8.3 ± 1.2 *	4.6 ± 2.7 *	1.8 ± 0.8 ^♦^
*AAI* _5_	-	0.032 ± 0.009	0.021 ± 0.004	-
*NA* _5_	-	3.5 ± 0.9	2.1 ± 1.6	-

* Indicates statistically significant difference (*p* < 0.05) in the values of *AAI* and *NA* for CLL patient groups compared with the control values; ^♦^ indicates statistically significant difference (*p* < 0.05) in the values of the *AAI* and *NA* for both patient groups receiving treatment compared with the respective values of the non-treated patients.

**Table 3 micromachines-16-00095-t003:** The ratio of unaggregated RBCs to the visible-field area, calculated for population 0 with an area <99 µm^2^, is as follows.

Healthy Controls	CLL Patients Without Treatment	CLL Patients Receiving Obinutuzumab/ Venetoclax	CLL Patients Receiving Ibrutinib Treatment
0.015 ± 0.002	0.0063 ± 0.001	0.0078 ± 0.001	0.014 ± 0.004

**Table 4 micromachines-16-00095-t004:** Percentage of RBC morphological types determined via optical microscopy of freshly isolated cells from healthy controls, untreated patients, patients receiving Ibrutinib, and patients receiving Obinutuzumab/Venetoclax.

Groups	RBC Morphological Types (%)		
	Biconcave	Echinocytes	Spherocytes
Healthy controls	83.3 ± 4.3	14.9 ± 3.7	1.8 ± 0.9
Untreated CLL patients	73.2 ± 6.6 *	21.2 ± 3.9 *	5.6 ± 2.0 *
CLL patients receiving Obinutuzumab/Venetoclax	71.7 ± 9.7 *	21.5 ± 8.9 *	6.8 ± 2.3 *
CLL patients receiving Ibrutinib treatment	81.4 ± 2.6	15.1 ± 4.5	3.5 ± 2.1

* Indicates a statistically significant difference (*p* > 0.05) in the percentage of RBC morphological types for CLL patient groups compared to healthy controls.

## Data Availability

All data are contained within the manuscript and available upon request.

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
