# Peer review of "The Impact of Targeted Therapies on Red Blood Cell Aggregation in Patients with Chronic Lymphocytic Leukemia Evaluated Using Software Image Flow Analysis"

_micromachines, 2025, doi:10.3390/mi16010095_

Round 1
Reviewer 1 Report
Comments and Suggestions for Authors
1. Line 105-106, why “the impact of current CLL treatments on red blood cell (RBC) rheology is of high interest”? Were there in vivo study of CLL treatments on red blood cell (RBC) rheology? Please give comments.
2. The “introduction” is too long, please make it brief.
3. Please give comments on the superiority of the microfluidic method used in this study compared to the above mentioned methods (Line 143-169).
4. Line 226-227, was the viscosity measured by rotated viscometer or tubulous viscometer? If it was rotated, please give the shear rate used. Furthermore, were the viscosities of all RBC suspensions 1.12mPaS? Though Hct and the concentration of Dex70 were the same, viscosity maybe different because rheological properties of RBC may affect the viscosity. Please give comments.
5. Line 317, please give the software of statistic and its version number. Normal distribution and homoscedasticity should be tested.
6. In table 1, WBC and lymphocytes in untreated group were not in “mean ±SD”.
7. The manuscript gave a new insight from rheology that seldom been mentioned. Please give comments on its possible application in clinics. The changes in the shape of RBCs are more easier to be observed.
Comments on the Quality of English Language
Though the English not very perfect, it can be understood.
Author Response
On behalf of all the co-authors, I would like to thank the reviewers for their valuable comments and suggestions. Detailed responses are provided below.
Comment 1. Line 105-106, why “the impact of current CLL treatments on red blood cell (RBC) rheology is of high interest”? Were there in vivo study of CLL treatments on red blood cell (RBC) rheology? Please give comments.
Response 1: We agree with the reviewer that the sentence written in this way does not accurately reflect the essence of the problem posed. In the revised version this inaccuracy has been corrected (lines 82 – 89): “There are still unresolved issues with the occurrence of side effects associated with bleeding, atrial fibrillation, development of hypertension, and further major cardiovascular events. These concerns may be, at least in part, attributed to downstream off-target effects, which can be linked to impaired microcirculation and altered rheological properties of RBCs [7,8]. Indeed, alterations in endothelial behavior have been observed in patients with lymphoproliferative disorders, including CLL [9, 10]. To shed light on this problem in the present study we characterized RBC rheology in CLL.”
Comment 2. The “introduction” is too long, please make it brief.
Response 2: In the revised version, the introduction has been shortened.
Comment 3. Please give comments on the superiority of the microfluidic method used in this study compared to the above mentioned methods (Line 143-169).
Response 3: Following the reviewer's recommendation, additional text has been added to highlight the advantages of the microfluidic method (second paragraph on page 2, and in the “Conclusion” section – the last paragraphs). Specifically, we have stated:
“The last decade witnessed remarkable progress in developing microfluidic devices, particularly for studying blood rheology. These devices offer several key advantages over traditional macroscopic rheometers, including using smaller sample volumes, achieving higher shear rates, and creating cost-effective, disposable devices with precise shear rates that mimic physiological conditions. These devices can also be combined with microscopy or other optical techniques to create a microfluidic system [18; 19] An important advantage of modern microfluidic systems is the achieved dimensions of the microfluidic channels, which can be considered as a model of blood vessels [20]. The microfluidic systems demonstrate high sensitivity in detecting small changes in RBC properties [21].”
“The Image Flow Analysis Software we have developed represents a powerful tool for both, basic research and clinical applications. It provides for a more nuanced and comprehensive view into RBC aggregation consisting of:
1) The ability to classify RBC aggregates based on their size and to capture subtle differences in their dimensions thus providing information on their contribution to various diseases;
2) Differentiation of aggregate types: The ability to distinguish between different aggregate types (such as rouleaux formations vs. 3D cluster networks) offers deeper insight into how aggregation mechanisms might change under pathological conditions.”
Comment 4. Line 226-227, was the viscosity measured by rotated viscometer or tubulous viscometer? If it was rotated, please give the shear rate used. Furthermore, were the viscosities of all RBC suspensions 1.12mPaS? Though Hct and the concentration of Dex70 were the same, viscosity maybe different because rheological properties of RBC may affect the viscosity. Please give comments.
Response 4: The viscosity was measured by a rotational viscometer (Brookfield DV-II+Pro).
The shear rate used was = 60.3s−1
We agree with the reviewer that hematocrit (Hct) is a key factor influencing blood viscosity in the body's microcirculation. However, in the experiment with RBC microfluidic suspensions, the Hct was set at up to 2%. As the Hct of RBC suspensions is very low (2%), meaning that it's a Newtonian fluid, and therefore the viscosity is constant for all samples and at all shear rates. Since the concentration of RBCs in this case is so low (2%), the rheological properties of the RBCs (e.g., aggregation, deformation) do not influence the overall viscosity of the fluid in a shear-dependent manner. [Stoltz, J.-F., Singh, M., Riha, P., 1999. Hemorheology in Practice. IOS Press, Netherlands, 1-116.; Thurston GB. Viscoelasticity of human blood. Biophys J. 1972 Sep;12(9):1205-17. doi: 10.1016/S0006-3495(72)86156-3].
Comment 5. Line 317, please give the software of statistic and its version number. Normal distribution and homoscedasticity should be tested.
Response 5: We thank the reviewer for the remark. The statistics software is now given in section 2.10. We used the Shapiro-Wilk test to determine the type of data distribution. As the results did not confirm a normal data distribution in all experimental groups, and the variances between the groups were not equal, we used a non-parametric test to obtain statistically valid results.
Comment 6. In table 1, WBC and lymphocytes in untreated group were not in “mean ±SD”.
Response 6: WBC and lymphocytes in the untreated group are not given as “mean ±SD” because the values vary significantly among the patients in this group. For this reason, we have given the lowest and highest values. In the revised version of the manuscript, this has now been reported below in Table 1.
Comment 7. The manuscript gave a new insight from rheology that seldom been mentioned. Please give comments on its possible application in clinics. The changes in the shape of RBCs are more easier to be observed.
Response 7: Our manuscript reveals the strength of microfluidic analysis combined with the newly proposed algorithm for Software Image Flow Analysis of RBC aggregates in characterizing the off-target effects of classical and newly developed therapies, specifically those affecting RBC aggregation and thus function. This is demonstrated by evidence that Ibrutinib preserves optimal microcirculation in CLL patients, whereas, the combination of Obinutuzumab/Venetoclax fails in this respect and therefore is associated with more severe side effects.
Many clinical treatments have subtle off-target effects that are not immediately apparent through standard blood tests. We show a proof of principle that monitoring RBC aggregation could provide a sensitive indicator of these effects, allowing: (1) the development of more precise therapies with fewer side effects on microcirculation, (2) patient stratification based on RBC aggregation as a complementary factor, (3) prompt adjustment of treatment by clinicians, well before more serious complications arise. Thus, our work is an important step towards forming personalized medicine protocols.
Comments on the Quality of English Language
Though the English not very perfect, it can be understood.
Response on the Quality of English Language: The text has been proofread by a professional English language translator.
Reviewer 2 Report
Comments and Suggestions for Authors
It is sad to admit that CLL “remains still incurable with conventional therapy.”
Therefore, any advances in the study of this disease are very valuable.
I welcome this work, but I recommend a major revision for the following reasons.
1. The authors should complete subsection 2.5. Microfluidic system description and experiments. I ask the authors to add a drawing of the microfluidic device with its dimensions and also provide a photograph of it. As I assume, the thickness of the channel roughly corresponds to the size of the red blood cell, so that the red blood cells form only one layer. I want to make this clear to the readers.
2. The authors introduce the term “the RBC aggregation index”.
Examples:
Line 300: AI = Si/Sv
Lines 449-451: “… the RBC aggregation index (AI), typically calculated as the ratio of the total area of all aggregates to the observed area in one visual field of the microscope.”
In my opinion, this term is not good enough. This may appear to be a physical characteristic of red blood cell aggregation in whole blood. However, the authors characterize the RBC suspension in fluid media at low HCT = 2%.
Line 219-220: “The final hematocrit of the RBC suspensions diluted in this way was reduced to 2% for all experiments.”
For sure, the RBC aggregation index will essentially depend on the hematocrit and adding of Dextran.
Authors should clarify this in the manuscript and show that their RBC aggregation index is a specific biological marker.
3. The authors consider AI1 (area 100–330 µm2) to be the minimum aggregate. I ask the authors to include in their Table 2 and graphical representation (Figures 2-5) AI0 (area < 99 µm2). These are individual disaggregated RBCs. I assume, that AI0 + AI1 + AI2 + AI3 + AI4 + AI5 » constant for all samples. It follows from the fact that HCT = 2% for all experiments and MCV is approximately the same for all groups (Table 1). Also, as I suggested in comment #1, RBCs only form one layer.
The constant AI0 + AI1 + AI2 + AI3 + AI4 + AI5 will confirm the accuracy of the experiment and the correctness of the image analysis.
4. The authors have repeatedly begun to discuss the mechanisms of erythrocyte aggregation (lines 535–538, 612–615, 622–629, 661–663, etc.). However, their findings are not specific.
The discussion is as follows: RBC aggregation depends on changes in RBC membrane characteristics. (Sorry for the oversimplification, but that's essentially it). I ask the authors to explain what characteristics, how they depend, for which groups (not only for this example).
5. Table 1, typo:
Change “MCV (fl)” to “MCV (fL)”
Author Response
It is sad to admit that CLL “remains still incurable with conventional therapy.”
Therefore, any advances in the study of this disease are very valuable.
I welcome this work, but I recommend a major revision for the following reasons.
Answer: On behalf of all the co-authors, I would like to thank the reviewers for their valuable comments and suggestions. Detailed responses are provided below.
Comment 1. The authors should complete subsection 2.5. Microfluidic system description and experiments. I ask the authors to add a drawing of the microfluidic device with its dimensions and also provide a photograph of it. As I assume, the thickness of the channel roughly corresponds to the size of the red blood cell, so that the red blood cells form only one layer. I want to make this clear to the readers.
Response 1: Following the reviewer's recommendation, we have added further details and references to Subsection 2.5 (and Supplementary Text 1 in Supplementary materials). In the Supplementary Materials, we have included photos of the BioFlux microfluidic system and the BioFlux 24-well plates (0-20 dyn/cm²), as well as a schematic diagram of the viewing area of a single microfluidic channel, which can be observed under an optical microscope. The microchannel has cross-sectional dimensions of 350 μm in width and 75 μm in thickness (as also described in Subsection 2.5, with further details available in the manufacturer's brochure, BioFlux Brochure, https://cellmicrosystems.com/wp-content/uploads/2024/09/BioFlux-Brochure_v6-Single-page.pdf). The scheme presented in Supplementary S2c illustrates the thickness of the microchannel (75 µm) and the direction of flow of the RBC suspension. Thus, the erythrocytes, with a diameter ranging from 6.2 to 8.2 μm and a maximum thickness of 2–2.5 μm, are smaller than the thickness of the microchannel. Therefore, when passing through the microchannel under flow conditions, the erythrocytes do not form a single layer but may move in multiple layers. At low flow rates (stasis or slow flow), reduced shear forces can allow the erythrocytes to aggregate and form structures such as rouleaux (a coin-like arrangement of stacked RBCs). These aggregates can further branch laterally or stack upon each other, leading to the formation of larger three-dimensional aggregates, as discussed in the manuscript.
A two-dimensional image is captured using the microscope, focusing on the most clearly visible layer from the multiple layers formed in the microchannel through which the RBC suspension flows. In this image, two- and three-dimensional aggregates, along with clusters of aggregates, can be observed. However, due to the technical limitations of the microscope, the depth of the three-dimensional aggregates cannot be analyzed.
Comment 2. The authors introduce the term “the RBC aggregation index”.
Examples:
Line 300: AI = Si/Sv
Lines 449-451: “… the RBC aggregation index (AI), typically calculated as the ratio of the total area of all aggregates to the observed area in one visual field of the microscope.”
In my opinion, this term is not good enough. This may appear to be a physical characteristic of red blood cell aggregation in whole blood. However, the authors characterize the RBC suspension in fluid media at low HCT = 2%.
Line 219-220: “The final hematocrit of the RBC suspensions diluted in this way was reduced to 2% for all experiments.”
For sure, the RBC aggregation index will essentially depend on the hematocrit and adding of Dextran.
Authors should clarify this in the manuscript and show that their RBC aggregation index is a specific biological marker.
Response 2: We thank the reviewer for his/her valuable feedback regarding the term "RBC aggregation index" (AI).
To address this, we have revised the term to more accurately reflect the context of our study. We replaced it with a more specific term "aggregation-area indicator" (AAI), to avoid confusion and better describe the measurement of RBC aggregation in the low hematocrit suspension. We also updated the corresponding sections in the manuscript.
We agree that the RBC aggregation index is likely to be influenced by factors such as hematocrit and the addition of Dextran, and we appreciate your suggestion to clarify this in the manuscript.
In this context, we have added a paragraph in the “Discussion” section highlighting the aggregation dynamics in a controlled low-hematocrit environment that is not directly applicable to whole blood samples, where hematocrit and other factors may have a more complex influence on aggregation and the presence of Dextran. We also emphasize that the RBC aggregation and its index, as used in our study, is specifically a biological marker for the aggregation behavior of RBCs in the low hematocrit suspension (HCT = 2%) and under the given experimental conditions, i.e., in presence of low concentration and low molecular weight of Dextran (last three paragraphs in a “Discussion” section). Specifically, we state:
“Our findings on the rheological changes in red blood cells, including their aggregation-area indicator, reflect their aggregation dynamics in a controlled environment with low hematocrit. However, in in vivo conditions, such as those found in whole blood, RBC aggregation is influenced by more complex factors, including variations in hematocrit and plasma protein levels, which can significantly affect aggregation in a more complex way. It is well known that RBC aggregation is particularly sensitive to hematocrit levels, especially when they exceed 20 - 30% [63]. As mentioned above, the bridging action of high-molecular-weight plasma proteins—such as fibrinogen and albumin—plays a critical role in RBC aggregate formation, particularly in larger veins [64]. Altered protein levels, influenced by various factors, such as inflammation, stress, and chronic diseases (e.g., diabetes, cardiovascular disease, cancer) can exacerbate the aggregation of RBCs and lead to higher blood viscosity, which in turn affects overall circulation [65, 66]. Therefore, RBC aggregation, deformability, and interaction with plasma proteins all contribute to blood viscosity and flow. Any changes or disturbances in these factors, caused by diseases or treatments, can lead to microcirculatory dysfunction, and impaired blood flow.
It is also to be noted that RBC aggregation increases in the presence of Dextran due to its macromolecular interactions with the RBC membrane, which alter the forces between RBCs, and promote the formation of rouleaux. The extent of aggregation depends on the concentration and molecular weight of Dextran. However, the concentration of Dextran used in our experiments is very low and equal across all samples, meaning its effect is minimal and does not influence the results obtained.
In the present work, we have excluded the influence of extrinsic factors, allowing us to specifically assess RBC aggregation based on their intrinsic biophysical characteristics and the changes induced by CLL and its corresponding treatment. Thus, our approach emphasizes the relationship between alterations in the biophysical properties of RBCs and their rheological behavior in disease. Additionally, RBC aggregation can be influenced by substances such as Dextran, a polysaccharide used to mimic blood viscosity and cell behavior. When introduced to a suspension of RBCs, Dextran acts as a macromolecular bridge that promotes RBC aggregation by facilitating intercellular interactions [67].“
Comment 3. The authors consider AI1 (area 100–330 µm2) to be the minimum aggregate. I ask the authors to include in their Table 2 and graphical representation (Figures 2-5) AI0 (area < 99 µm2). These are individual disaggregated RBCs. I assume, that AI0 + AI1 + AI2 + AI3 + AI4 + AI5 » constant for all samples. It follows from the fact that HCT = 2% for all experiments and MCV is approximately the same for all groups (Table 1). Also, as I suggested in comment #1, RBCs only form one layer.
The constant AI0 + AI1 + AI2 + AI3 + AI4 + AI5 will confirm the accuracy of the experiment and the correctness of the image analysis.
Response 3: Hematocrit refers to the percentage of blood volume occupied by red blood cells. In our BioFlux image analysis, we calculated the area of rouleaux and more complex (3D) aggregates. As mentioned previously, the RBCs in these aggregates often overlap in several layers. Due to the technical limitations of the microscope, we were unable to capture the depth of these three-dimensional aggregates. Therefore, we measured only the visible area of the aggregates, as explained in the response to Comment 1.
As a result, directly comparing the sum of the areas of these aggregates with hematocrit is not accurate as area measurements do not correlate with the volume of red blood cells in the entire sample. The formation of aggregates is influenced by various factors such as cell-cell interactions, the biophysical properties of RBCs, and changes in these properties under different conditions, including pathological states. These complexities make it difficult to draw a straightforward correlation between aggregate area and hematocrit values. To accurately assess RBC volume in the samples, other methods that account for both size and 3-D structural characteristics of RBCs, such as volume-based imaging techniques would be more appropriate.
We appreciate your suggestion to include an additional population (Population 0) representing individual disaggregated RBCs (with area <99 µm²) in our analysis. Following your recommendation, we have depicted the RBCs from Population 0 in Figures 2 to 4.
However, since AAI refers to parameters that, specifically reflect aggregation behavior, we did not include AAI0 in Table 2. Instead, we created an additional table (Table 3 in the revised version), where we present the ratio of the area of unaggregated RBCs to the visible field area, specifically for Population 0 (with area < 99 µm²). This was done to provide a more accurate representation of this subpopulation and to ensure that the reader understands the distinction between the populations and the purpose of Table 3.
For the same reason, we believe it would not be appropriate to include this parameter in Figure 5, as it is not directly related to aggregation indices.
We agree with your assumption that, under the conditions of HCT = 2% and a consistent MCV across all groups, the sum of AI0 + AI1 + AI2 + AI3 + AI4 + AI5 should remain approximately constant for all samples. However, this is only true if the volume of the aggregates can be measured, which is not applicable in our experiments.
Nevertheless, based on the results presented in Table 3, we discuss the relationship between these components in the manuscript (the last paragraph of the “Results:” section). Specifically, we state: “We also observed that the ratio of unaggregated erythrocytes to the visible field area, calculated for Population 0 (with an area <99 µm²), was the highest in the healthy group and CLL patients treated with Ibrutinib (Table 3). A significantly lower value for this parameter was found in untreated CLL patients and those receiving Obinutuzumab/Venetoclax. This decrease of unaggregated RBCs in the latter groups provides further evidence of increased RBC aggregation in these patients.”.
Comment 4. The authors have repeatedly begun to discuss the mechanisms of erythrocyte aggregation (lines 535–538, 612–615, 622–629, 661–663, etc.). However, their findings are not specific.
The discussion is as follows: RBC aggregation depends on changes in RBC membrane characteristics. (Sorry for the oversimplification, but that's essentially it). I ask the authors to explain what characteristics, how they depend, and for which groups (not only for this example).
Response 4: Thank you for your thoughtful comment. We agree that the current explanations could benefit from further specificity and clarity, particularly regarding the specific characteristics of RBC membrane changes and how they influence aggregation.
We have revised the discussion to precisely, identify the key factors contributing to RBC aggregation. Specifically, we have elaborated on how changes in the RBC membrane characteristics—such as membrane fluidity, shape, and composition of its protein and lipid membrane, and cytoskeletal integrity —affect aggregation behavior (the fourth and fifth paragraph of 4.1. Alterations of RBC rheological properties in untreated CLL patients).:
“RBC deformability is largely determined by the elasticity and flexibility of their mem-brane, affected by a complex interplay of factors, including the lipid and protein composition of the membrane, cytoskeletal integrity, environmental conditions (such as pH, ion concentrations, etc.), and genetic factors. Diseases and treatments that disrupt any of these factors can significantly impair RBC deformability. Key membrane proteins like spectrin, ankyrin, and band 3 are essential for maintaining the structural integrity of the RBC membrane. For example, it has been shown that a reduction in the number of ankyrin-binding sites in the cytoplasmic domain of band 3 in red blood cells from patients with chronic myeloid leukemia partially disrupts the connection between the cytoskeleton and the membrane [36]. In hereditary spherocytosis, the deficiency or dysfunction of proteins like α-spectrin, β-spectrin, ankyrin, band 3, or protein 4.2 impairs the normal interaction between the RBC membrane and its cytoskeletal network, making the RBCs more fragile and prone to hemolysis. “
“Another influencing factor, such as the aging of RBCs leads to increased aggregation in microcirculation. In their work, Puthumana and colleagues state that aging and increased cell density are associated with dehydration, membrane loss, and other changes that lead to smaller, less deformable, and more spherical cells (e.g., echinocytes), and consequently to increased RBC aggregation [43]. In line with this statement, we found a higher proportion of echinocytes and spherocytes in the samples of untreated CLL patients and those receiving Obinutuzumab and Venetoclax evidencing an accelerated RBC aging in these patients as mentioned previously.”
We have also clarified how the cell’s aggregation depends on the experimental conditions, including the presence of specific aggregating agents (like Dextran), and hematocrit (according to Comment 2).
Comment 5. Table 1, typo:
Change “MCV (fl)” to “MCV (fL)”
Response 5: Now it is corrected.
Round 2
Reviewer 2 Report
Comments and Suggestions for Authors
The authors made notable additions to the manuscript in response to my comments. In my opinion, the manuscript is now significantly improved and can be recommended for publication.